# Health Benefits of Antioxidative Peptides Derived from Legume Proteins with a High Amino Acid Score

**DOI:** 10.3390/antiox10020316

**Published:** 2021-02-20

**Authors:** Athanasia Matemu, Soichiro Nakamura, Shigeru Katayama

**Affiliations:** 1Department of Food Biotechnology and Nutritional Sciences, Nelson Mandela African Institution of Science and Technology, Arusha P.O. Box 447, Tanzania; athanasia.matemu@nm-aist.ac.tz; 2Graduate School of Science and Technology, Shinshu University, 8304 Minamiminowa, Kamiina, Nagano 399-4598, Japan; snakamu@shinshu-u.ac.jp; 3Institute for Biomedical Sciences, Shinshu University, 8304 Minamiminowa, Kamiina, Nagano 399-4598, Japan

**Keywords:** legume, soybean, chickpea, lentil, cowpea, mung bean, antioxidant peptides, health benefits, amino acids

## Abstract

Legumes such as soybean, chickpea, lentil, cowpea, and mung bean, are valuable sources of protein with a high amino acid score and can provide bioactive peptides. This manuscript presents a review on legume-derived peptides, focusing on in vitro and in vivo studies on the potential antioxidative activities of protein hydrolysates and their characterization, amino acid sequences, or purified/novel peptides. The health implications of legume-derived antioxidative peptides in reducing the risks of cancer and cardiovascular diseases are linked with their potent action against oxidation and inflammation. The molecular weight profiles and amino acid sequences of purified and characterized legume-derived antioxidant peptides are not well established. Therefore, further exploration of legume protein hydrolysates is necessary for assessing the potential applications of antioxidant-derived peptides in the functional food industry.

## 1. Introduction

Legumes are of global economic importance and provide 33% of dietary plant proteins to millions of people [1]. Legumes are rich in protein (16–50%), vitamins, minerals and bioactive compounds. Legumes are also known as a source of natural antioxidants [2]. Nutritionally, soybean, chickpea, mung bean, cowpea and lentil are the important grain legumes in Asian and African countries [3]. The dietary quality of some legumes is attributed to the high-quality proteins and peptides and the well-balanced essential amino acids they contain. Their consumption reportedly reduces the risk of chronic diseases. Legumes are a cheaper source of proteins compared to the more expensive animal proteins that are consumed in most parts of the world. Advances in functional food science have resulted in food and food ingredients with diverse applications, including free radical-scavenging. Furthermore, innovation in the field of functional foods and nutraceuticals has directed the development of effective natural antioxidants from legume-derived proteins. With the currently growing demand for healthier foods, products that can scavenge and quench free reactive oxygen species (ROS)/radicals, stabilize them and maximize health effects have attracted more attention. A growing interest in natural antioxidants has led to intensive research on biofunctional peptides derived from a wide variety of legume-based food products and byproducts. These biofunctional peptides are encoded in an inactive form in the protein structure and can be produced by digestive enzymes during gastrointestinal digestion as well as nongastrointestinal digestion using commercial proteases and fermentation.

During the last few decades, natural antioxidants from legume-derived proteins such as soybean, chickpea, lentil, mung bean, and cowpea have attracted considerable interest as safer alternatives to synthetic antioxidants for restoring the oxidant–antioxidant balance, thereby reducing the risk of oxidative-stress related diseases [4,5]. An imbalance between ROS and reactive nitrogen species (RNS) and the antioxidant defense mechanism [6] can lead to the development and progression of many diseases [7,8,9,10]. The antioxidant defense mechanism is determined by the hydrogen-donating, electron-donating, metal-ion-chelating, and radical-scavenging abilities of the resultant peptides [11,12,13,14]. In addition to their nutritional benefits, legume-derived hydrolysates and peptides can exhibit various biofunctions such as antioxidant, hypolipidemic, antihypertensive, anticancer, anti-inflammatory, and immunomodulatory effects [15,16,17,18]. Since peptide size and sequence are important criteria that determine the antioxidative property of protein hydrolysates and identify the most effective peptides can provide insights into the underlying mechanism of action, thereby ascertaining their structure–function claims. The exploration of active legume-derived peptides and the identification of their sequences using either analytical techniques or bioinformatics tools is essential for the functional food industry. This review focuses on the characterization and health impact of antioxidative legume-derived hydrolysates/peptides and their sequences, as influenced by their amino acid composition.

## 2. Antioxidant Properties of Individual Amino Acids

All amino acids have potential antioxidant properties [19]. A good balance among the various amino acids is essential for the antioxidative properties of plant-based proteins. The nutritional quality of plant-based proteins is lower than that of animal-based proteins because of deficient or unbalanced essential amino acids, particularly Lys in cereals and nuts [6,20,21,22]. Nevertheless, soybean [23], chickpea [24], lentil [25,26,27], cowpea [25,26], and mung bean [25] are good sources of high-quality proteins and fulfill the requirement for Lys. The most physiologically active peptides have a low molecular weight and short sequences of 2–20 amino acids [28,29]. Lentil and mung bean proteins are low in sulfur-containing amino acids such as Met and Cys [25]; however, they have a good balance of other amino acids and resultantly can exhibit high antioxidant activity.

Some plant-based food products that have a specifically high total protein concentration contain roughly equivalent total indispensable amino acid levels, whereas some can transfer electrons, thereby exhibiting antioxidant activity [30]. A strong antioxidant activity can be attributed to the unique composition, sequence, size, and characteristics of free amino acid side chains [6,31]. Plant-derived peptides may exert antioxidant activity via certain amino acids acting as metal-chelating and hydrogen-/electron-donating agents through interactions with free radicals, thereby terminating the radical chain reaction or preventing their formation [32,33]. Nonetheless, the presence of aromatic (Try, Tyr and Phe), imidazole (His), and sulfur-containing amino (Cys and Met) groups [31,33] in peptides results in the development of antioxidative properties. Several individual amino acids, such as Tyr, Met, His, Lys, Pro, and Trp, reportedly influence antioxidative activity or are accepted as antioxidants and can be predetermined by their amino acid sequence and composition [16,34]. Soybean protein hydrolysate fractions with significantly higher amounts of His, Tyr, and Phe exhibit better radical-scavenging activity than the other fractions [35]. For instance, Trp amino acid residues can considerably scavenge free radicals to break chain reactions via the donation of the hydrogen attached to the nitrogen of its indole ring [28]. Specifically, antioxidant properties are facilitated by His-containing peptides with metal-ion chelation, active oxygen-quenching [16], and free radical-scavenging properties [17,36]. In another study, the antioxidant activity of Asp, Glu, His, Arg, and Tyr in a 717.37 Da novel chickpea peptide was largely related to its imidazole ring from the histidine residue. Additionally, acidic and/or basic amino acids play important roles in the chelation of metal ions by their functional groups [34]. Similarly, the presence of Tyr, Met, His, Lys, and Trp has been shown to influence the antioxidant activity of soy peptides [11,16,17,37]. Therefore, individual amino acids play an important role in the antioxidant property of the peptide fraction. Specifically, bioactive peptides show antioxidant activities, which are attributed to their radical-scavenging, lipid peroxidation-inhibiting, and metal-ion-chelating properties [38].

## 3. Production, Characterization, and Health Impact of Soybean-Derived Antioxidant Peptides

Soybean (*Glycine max*) is a rich source of high-quality proteins and natural bioactive peptides, with potential for health promotion and reduction of the risk of chronic diseases. The nature of the process has a significant impact on the biofunctionality of peptides; therefore, proper selection of a method for the generation of active amino acids based on size, charge, and hydrophobicity is vital [6]. Various methods such as enzymatic hydrolysis [39,40], gastrointestinal digestion [39], microbial fermentation [6,12,39,41,42], germination [13,18,43,44], chemical modification [15,17], and membrane filtration [37,45,46,47] are employed to produce soy peptides or their mixtures. These methods help to produce enormous amounts of small peptides with antioxidant properties from soy protein [11,40]. Radical-scavenging activities can be increased three- to fivefold through enzymatic digestion [15]. Similarly, Nwachukwu and Aluko reported the considerable hydroxyl radical-scavenging ability of soybean protein hydrolysates and peptide fractions with a higher degree of hydrolysis (low molecular weight) [37]. In addition, the enhanced effect of fermented soybean meal, soy foods, and soymilk [41,48,49,50] against oxidative stress has been reported.

Scientific reports have shown that most antioxidative properties are associated with uncharacterized soy protein hydrolysates (SPH) or products (Table 1). This further suggests the importance of characterizing potent protein hydrolysates to generate active peptides with a specific size, molecular weight, amino acid composition, and sequence. Thus, exhaustive peptide characterization would allow for the identification and purification of the peptide fractions responsible for their activity and optimize their production [32]. Table 1 shows the various methods used for the preparation and characterization of antioxidant SPH and peptides. The latest development on the characterization of antioxidative peptides has focused on advanced analytical techniques for their extraction, identification, and quantification [15,17,51]. According to Piovesana et al., these analytical techniques would greatly benefit the discovery of new biofunctional peptides [29]. Liquid chromatography–electrospray ionization mass spectrometry (LC-ESI-MS/MS) and matrix-assisted laser desorption/ionization time of flight/time of flight (MALDI-TOF/TOF) mass spectrometry are some of the powerful techniques for the characterization and quantification of proteoforms [11,45,52,53]. The biofunctional and health implications of antioxidative plant-derived peptides in reducing the risks of noncommunicable diseases such as cancer and cardiovascular diseases are linked with their potent action against oxidation and inflammation [54].

Plant protein-derived compounds can play a key role in combating metabolic diseases and age-related disorders resulting from oxidative stress. Oxidative stress is associated with various pathological conditions including type-II diabetes, neurodegenerative diseases, immunosuppression, cancer (certain types), early senescence, obesity, and cardiometabolic disorders [15,65,66,67]. With growing research interest in functional food science, soy products have attracted attention as safer alternatives for therapeutic applications. Soy intake has been linked with the reduction of the risk of various chronic diseases [68], via beneficial effects exerted through interactions with cell receptors, through functions as hormones and regulating enzymes, or by interfering with cell cycles [17]. Evidence from epidemiological studies links the consumption of high levels of soy products with low incidence of cancer [17]. Furthermore, soybean peptides modulate immune function, brain function, and neurochemistry in humans [69]. For instance, a health claim on the ability of soy proteins to reduce the risk of coronary heart disease has been approved in the USA [15]. This further substantiates the health-related importance of soy products and their roles in the prevention of chronic diseases. Interestingly, some soy peptides such as lunasin [a 43-amino acid peptide (SKWQHQQDSCRKQKQGVNLTPCEKHIMEKIQGRGDDDDDDDDD)] and soymorphins [a five-amino-acid (YPFVV) peptide] have been reported to possess protective effects against cancer [16] and cardiovascular diseases [4] and induce improved glucose and lipid metabolism in diabetic KKAy mice [70]. Studies have demonstrated the ability of different soy products such as soy sauce to reduce oxidative damage and the role of such products in disease prevention [13,18,65]. A number of soybean hydrolysates or peptides have been reported to prevent oxidative stress in intestinal Caco-2 cells [4], the liposomal system [36], HL-7702 human hepatocyte cells [13], human astrocyte U373MG cells [65], HeLa cells, and in *Caenorhabditis elegans* [52].

In vitro studies have also demonstrated the cancer-preventive effect of the peptide lunasin in RAW 264.7 macrophages [54] and germinated peptide fraction (>10 kDa) in human cervical carcinoma (HeLa, SiHa, CasKi) and human breast (MCF-7 and MDA-MB-231) cancer cells, respectively [18], and that of fermented soybean in MCF-7 [71]. The anticancer property of lunasin is reportedly achieved through its activity against oxidation and inflammation, which are the causative factors for carcinogenesis [54]. Specifically, lunasin, a powerful and naturally occurring antioxidant, inhibits ABTS, ROS production, and the release of proinflammatory cytokines (TNF-α and IL-6) by scavenging peroxyl and superoxide radicals; it also prevents ROS generation and the glutathione peroxidase and catalase activities in vitro [39,54] (Table 2). Likewise, soymetide, a 13-amino acid peptide (MITLAIPVNKPGR), exhibits immunomodulation activity in humans [72]. In contrast, the peptide fraction (>10 kDa) is the most potent anticancer agent influenced by antioxidative aromatic amino acids such as Tyr and Phe with a low molecular size.

Hypocholesterolemic peptides from SPH (IAVPGEVA, IAVPTGVA, and LPYP) can interfere with the catalytic activity of 3-hydroxy-3-methylglutaryl CoA reductase (HMGCoAR), one of the main enzymes in cholesterol biosynthesis, and modulate cholesterol metabolism by activating the LDLR-SREBP2 pathway, consequently increasing the ability of low-density lipoprotein (LDL) uptake by HepG2 cells [73]. Immunomodulatory peptides are important in delaying the progression of degenerative diseases such as cancer, thereby preventing its incidence. A study by Ali et al. has demonstrated that fermented soybean stimulates splenocyte cytokine (IL-2 and IFN-γ) production [71]. This further justifies the potential of soy peptides in preventing or reducing the risk of degenerative diseases. As both anti-inflammatory and antioxidant properties are the key indicators of the etiology of cancer prevention, these properties can be considered as predetermined when searching for anticancer agents in legume proteins.

## 4. Production, Characterization, and Health Impact of Chickpea-Derived Antioxidant Peptides

Chickpea (*Cicer arietinum* L.) is a cheap source of dietary protein (15–30%) [1,24,26,74,75,76] with a well-balanced essential amino acid composition. Chickpea is the most important pulse crop and is the second most widely produced crop worldwide [1]. Chickpea contains acceptable levels of essential amino acids, except for the sulfur-containing amino acids His, Tyr, Met, and Cys [24,26,77,78]. Various methods have been employed to produce chickpea protein hydrolysates with improved biofunctional properties such as antioxidant, hypocholesterolemic, and angiotensin I-converting enzyme (ACE) inhibitory activity; metal-chelating ability; and antihyperlipidemic, antitumor, and antiproliferative effects [19,33,74,79,80,81,82,83,84,85,86]. The purification of chickpea protein hydrolysates using the chromatography technique (affinity and size) generates small metal-chelating peptide fractions with 1 to 11 amino acid residues rich in His (20–30%) [83]. Chickpea protein hydrolysate fractions RQSHFANAQP (1155 Da) [33] and NFYHE (717.37 Da) [34] have been shown to exhibit the highest antioxidant activity (Table 2). According to Hernandez et al. [87], in humans, supporting evidence for only two chickpea peptides, NFYHE and RQSHFANAQP, as potential antioxidants exist. Moreover, a synthetic version of RQSHFANAQP exhibits antioxidant activity and antiproliferative effect both in vitro and in vivo [85]. Torres-Fuentes et al. [32] further demonstrated four antioxidant chickpea peptides rich in His, Tyr, and Phe, which are ALEPDHR, TETWNPNHPEL, FVPH, and SAEHGSLH (Table 2). Similarly, reverse phase-high-performance liquid chromatography coupled with tandem mass spectrometry (RP-HPLC-MS/MS), LC-ESI-MS/MS, and MALDI-TOF were employed to isolate novel peptide fractions of defined hydrophobicity, size, and net charge within the most active peptides [32,33]. Recently, a number of novel antioxidant peptides from chickpea, DHG (327.33 Da), VGDI (402.49 Da), [74] and LTEIIP (685.41 Da), have been isolated [74,88]. Notably, the highest antioxidant and free radical-scavenging activities of chickpea hydrolysate fractions rich in Arg, Phe, Lys, Leu, Ala, and Asp with a low molecular size (200–3000 Da) and high hydrophobicity score have also been reported [34,80] (Table 2). Of note, in one way or another, all active tri-, tetra-, penta-, and oligopeptides have been characterized by low molecular weight/size, a unique amino acid composition and sequence, hydrophobicity, and the N- or C- terminal amino acid. Additionally, these structural properties are specific to the antioxidative peptides and are necessary to inhibit degenerative diseases, thereby having health-promoting effects [28,89].

## 5. Production, Characterization, and Health Impact of Lentil-Derived Antioxidant Peptides

Lentils (Lens culinaris) are important leguminous crops that are consumed in more than 100 countries worldwide; these are known to exert marked health benefits. The global production of lentils has significantly increased in the last five decades [91]. The protein content of lentils ranges between 20−30% [26,48] (legumins (~45%), albumin (~17%), glutelins (~11%), vicilin (~4%), and prolamins (~3%)) [1]. Lentils are rich in Lys, Arg, and Leu but deficient in Met, Trp, and Cys [26]. Structural modification is inevitable for releasing biofunctional peptides present in lentil proteins. Enzymatic hydrolysis, germination, and fermentation are some of the treatments used to release biofunctional peptides from lentils [44,91,92,93,94]. For instance, spouting significantly impacts the levels of essential amino acids (Ile, Leu, Lys, total aromatic amino acids, and Try, but not that of total sulfur amino acids) and the protein efficiency ratio (PER) of lentils [44]. As in the case of other leguminous proteins, the antioxidative property of lentils is reflected in non-purified hydrolysates, extracts, or fermented products (Table 3). Fractionation of the protein hydrolysate can reveal a direct relationship between the structure and functional activities of the proteins/peptides [28]. Likewise, the characterization of potent biofunctional peptides from leguminous proteins is still limited due to peptide mixtures [95]. Therefore, the identification of active peptide sequences is needed for the elucidation of their activities against chronic disorders. Lentil protein hydrolysates and peptide fractions with diverse functional benefits such as antioxidant, ACE inhibitory, antihypertensive, and antidiabetic activities have been reported [91,92,95] (Table 3). Lentils fermented by Bacillus subtilis have been reported to raise oxygen radical absorbance capacity (ORAC) levels from 0.17 to 0.22 and 0.24 mmol Trolox equivalent (TE)/g for up to 96 h [92]. Furthermore, multifunctional lentil ingredients targeted for metabolic syndrome (MetS) were generated by subjecting it to the activity of *Lactobacillus plantarum* CECT 748 combined with savinase-hydrolysis (LPHS). Additionally, LC-MS/MS-purified LPHS-F1 peptides (SDQENPFIFK, HGDPEER, and HGDPEER) exhibited high ORAC values and the inhibition of ROS generation in RAW 264.7 macrophages [93].

Similarly, RP-HPLC MS-purified LLSGTQNQPSFLSGF, NSLTLPILRYL, and TLEPNSVFLPVLLH from lentil vicilin, convicilin, and legumin, respectively, exhibited the highest antioxidative activity (0.013–1.432 μmol TE/μmol peptide), with the C-terminal heptapeptide being the crucial factor for the observed activity. Nevertheless, these peptides were also effective inhibitors of ACE activity (IC_50_ = 44−120 µM) [95].

The nutritional characteristics of lentils have been linked with the reduction in the incidence of various cancers and type-II diabetes due to their antioxidant and anti-inflammatory potentials. The LPHS-F1 peptides SDQENPFIFK, HGDPEER, and ATAFGLMK (838–1225 Da) demonstrated in vitro multifunctional roles as strong oxygen radical-scavenging, antihypertensive, and hypoglycemic agents. Notably, ATAFGLMK generated from ADP-glucose pyrophosphorylase, a key regulatory enzyme for starch biosynthesis, displayed both antioxidant and inhibitory activities against glucose synthesis [93]. Thus, the structural characteristics displayed by the peptides further postulate their multifunctional roles. In a recent study by Evcan et al. [96], hydrolyzed lentil protein–iron complexes (10:1) significantly reduced the anemic condition in CaCo-2 cells by reducing the mRNA levels of iron-regulated divalent metal transporter-1 (DMT1), transferrin receptor (TFR), and ankyrin repeat domain 37 (ANKRD37) marker genes. A naturally occurring lentil BBI inhibited the proliferation of human colon adenocarcinoma HT29 cells in a dose-dependent manner and at concentrations > 19 µM, but not affecting the colonic fibroblast CCD-18Co cells [97].

## 6. Production, Characterization, and Health Impact of Cowpea-Derived Antioxidant Peptides

Cowpea (*Vigna unguiculata*) is a legume consumed in many parts of the world, with protein content ranging from 22–30% [1,98]. Cowpea contains high-quality protein with high digestibility and is composed of globulins (50–70%) as its 11S (legumin) and 7S (vicilin/β-vignina) fractions [1,99]. Lys is the most abundant amino acid, with Leu, Val, and Phe being slightly higher than sulfur-containing amino acids. Upon hydrolysis, cowpea releases biofunctional peptides with antioxidant [100,101,102], antihypertensive [102], anti-inflammatory, hyperglycemic [103] and hypocholesterolemic [102] activities (Table 4). Germination and enzymatic hydrolysis of cowpea bean generated TTAGLLE capable of inhibiting dipeptidyl peptidase IV [104]. Further, the antioxidant activity of peptide fractions can be obtained by the ultrafiltration of cowpea Flavourzyme hydrolysates (<1 kDa) [101]. Like other legumes, cowpea protein hydrolysates are not well-characterized, despite their well-known biofunctional properties. Therefore, biofunctional peptide profiles and the amino acid sequence of cowpea hydrolysates are necessary to optimize their applications to benefit human health [98].

Cowpea has been associated with the prevention of cardiovascular disease, managing type-II diabetes, colon cancer, and lowering LDL cholesterol levels [22]. The hypocholesterolemic GCTLN peptides obtained by in vitro-simulated human digestion of cowpea bean proteins inhibit the initial 3-hydroxy-3-methylglutaryl coenzyme A (HMG-CoA) reductase activity from 47.8% to 57.1% through changes in its active site [102]. Cowpea peptides display insulin resistance to Rat L6 skeletal muscle cells by inducing Akt phosphorylation in the cell culture, thereby activating the insulin signaling cascade. It has been further shown that cowpea peptides can mimic the actions of insulin [103]. In another study, cowpea peptide (<3 kDa) inhibited the enzyme HMGCR and reduced cholesterol micellar solubilization in vitro [105]. The peptic and tryptic hydrolysates of cowpea seed protein showed concentration-dependent inhibition of α-amylase in converting starch to maltose, by 82.86% and 74.91%, respectively [106]. The cowpea hydrolysates (≤3 kDa) modulated the gene expression of cholesterol transporters by altering NPC1L1, ABCA1, and ABCG1 mRNA levels in Caco-2 cells [102].

**Table 4 antioxidants-10-00316-t004:** Antioxidant properties of cowpea-derived hydrolysate.

Source of Peptides	Preparation Method/s	Antioxidative Cowpea Peptides	Antioxidant Properties (in vitro or in vivo)	Reference
Cowpea seeds	Alcalase, Flavourzyme, Pepsin, pancreatin, UF	<1 kDa hydrolysate	ABTS	[101]
Cowpea protein	Pepsin, pancreatin, UF	>1 kDa hydrolysate	FRAP, ORAC, SRSA	[107]
Cowpea isolate	Pepsin	Hydrolysates	DPPH	[100]
Cowpea seed protein	Pepsin, trypsin	Hydrolysates	H_2_O_2_ scavenging, FRAP	[106]

## 7. Production, Characterization, and Health Impact of Mung Bean-Derived Antioxidant Peptides

Mung bean (*Vigna radiata*), also known as green gram or moong bean, is consumed worldwide. Mung bean is good source of protein (19.5–33.1%) [108] and peptides, making it a functional food and an excellent dietary source of antioxidants [109]. Mung bean protein is rich in Phe, Try, Leu, and Lys but deficient in the sulfur-containing amino acids Met, Cys, and Trp [110]. A list of mung bean protein hydrolysates and peptides with antioxidative activities is shown in Table 5. LC-MS-purified peptide F37 (855–1308 Da) hydrolyzed by *Virgibacillus* sp. SK37 proteinase with four peptides containing an Arg residue at their C-terminus exerts the highest specific antioxidant activity. Likewise, purified peptide F42 (775–1233 Da) demonstrates ABTS radical-scavenging activity (0.148 mg/mL) similar to BHT control (Table 5). Try is found at the C-terminus as a hydrogen donor in mung bean peptides [111]. Ultrafiltered-Alcalase mung bean protein hydrolysate-I (MBPH-I) presents the best scavenging DPPH, hydroxyl radical, superoxide radical, and Fe^2+^-chelating activities in addition to the best ACE inhibitory activity (IC_50_ = 4.66 μg/mL) [112].

Mung bean is rich in biofunctional peptides capable of providing health benefits to humans such antihypertension and anti-inflammation. YADLVE (708.33 Da) strongly inhibited renin activity at 97% in vitro in the presence of Leu (hydrophobic amino acid) within peptide sequence contributed to renin inhibitions. According to Dianzhi Hou et al. [113], small mung bean peptides with hydrophobic amino acid residues show higher bioactivity. Additionally, YADLVE also reduces blood pressure to up to 36 mmHg over a 24 h period after it is administered orally to spontaneously hypertensive rats [114]. Mung bean protein hydrolysate exerts strong and dose-dependent suppressing effects on proinflammatory mediators by blocking the nuclear factor-kappa B (NF-κB) pathway in liposaccharide (LPS)-stimulated RAW 264.7 macrophages [115]. In contrast, MBPH-I (<3 kDa) effectively protects NCTC-1469 cells from damage caused by free radicals (H_2_O_2_-induced cell injury) and is influenced by high contents of hydrophobic amino acids and aromatic amino acids. Additional MBPH displays excellent ACE inhibitory activity (IC_50_ = 4.66 μg/mL) in vitro. Thus, the multifunctional properties of MBPH-I suggest its potential application as a natural functional ingredient [108]. In vivo antioxidant and hepatoprotective effects of aqueous extracts from germinated and fermented mung beans enriched with amino acids and γ-aminobutyric acid (GABA) on ethanol-induced hepatoxicity in mice have been reported. The antioxidant properties of the extracts were examined in mice liver tissue via malondialdehyde (MDA), superoxide dismutase (SOD), ferric-reducing antioxidant power (FRAP), and nitric oxide (NO). Mice treated with fermented and germinated mung bean extract show hepatoprotective potential through an increase in the SOD and FRAP levels, decrease in MDA, and suppression of NO production [116]. In another in vivo study, fermented mung bean extracts significantly reduced the blood sugar levels of mice with glucose-induced diabetes. Similarly, mung beans fermented by Mardi *Rhizopus* sp. strain 5351 also enhance the antihyperglycemic and antioxidant effects in alloxan-treated mice. The hyperglycemic effect occurs through the lowering of the serum levels of cholesterol, triglyceride (TG), and low-density lipoprotein (LDL), whereas insulin secretion and the antioxidant level for MDA are significantly improved in the plasma of the fermented mung bean treated group in alloxan-induced hyperglycemic mice [117]. Moreover, fermented mung beans with a high concentration of amino acids show the highest cytotoxicity activity against breast cancer MCF-7 cells by arresting the G0/G1 phase followed by apoptosis. Furthermore, fermented mung beans have immunostimulatory effects on mouse splenocytes by inducing splenocyte proliferation and enhancing the levels of serum IL-2 and IFN-γ. The presence of free amino acids among others has been previously reported to enhance cytotoxicity and immunomodulation [71]. Mung bean protein hydrolysate has been shown to alleviate H_2_O_2_ (50 μM, 30 min) genotoxicity on hepatoblastoma HepG2 [118] as well. Interestingly, both intact mung bean vicilin protein (MBVP) and Alcalase- and trypsin-generated mung bean vicilin protein hydrolysates (AMBVPH and TMBVPH) show dose-dependent antiproliferative activities against MDA-MB-231 and MCF-7 breast cancer cells in addition to antioxidant and ACE inhibitory activities [119].

**Table 5 antioxidants-10-00316-t005:** Antioxidant properties of mung bean-derived hydrolysates and peptides.

Source of Peptides	Preparation Method/s	Antioxidative Mung Bean Peptides	Antioxidant Properties (in vitro or in vivo)	Reference
Mung bean meal	*Virgibacillus* sp. SK37 proteinases, Alcalase, Neutrase, UF, IEC, SEC	FLGSFLYEYSR (1380.6Da), AVKPEPAR (866.5 Da), GVGLFVR (746.4 Da), HNVAMER (855.4 Da), LGSFLYEYSR (1233.6Da), LLPHLR (903.5Da), FNVPATK (775.4Da), SGVVPGY (677.3 Da)	ABTS, FRAP, Fe^2+^ chelation	[111]
Mung bean protein	UF, Alcalase, Neutrase, Protamex, papain	Fraction I (<3 kDa), II (3–10 kDa), III (>10 kDa)	DPPH, HRSA, Fe^2+^ chelation, SRSA, reducing power	[112]
Mung bean	Mardi *Rhizopus* sp. strain 5351	Fermentates	MDA in alloxan-treated mice	[117]
Mung bean	Alcalase, UF	Fraction I (<3 kDa), II (3–10 kDa), III (>10 kDa)	MDA, GSH, SOD, and LDH in injured NCTC-1469 cells, inhibition of ROS generation in NCTC-1469 cells	[108]
Mung bean protein	Ficain, Bromelain	Hydrolysates	DPPH, Fe^2+^ chelation, inhibition of lipid oxidation	[120]
Mung bean	Germination, *Rhizopus* sp. strain 5351	Germinated and fermented mung bean aqueous extracts	SOD, MDA, FRAP, NO	[116]
Mung bean seedling	IEC, UF, HPLC	Low molecular weight peptides (0.5–3 kDa, 3–10 kDa)	DPPH, ABTS	[121]
Mung bean	*Rhizopus* sp. strain 5351	Fermentates	DPPH, FRAP	[71]
Mung bean vicilin protein	Alcalase, trypsin, IEC	Hydrolysates	DPPH, ABTS, FRAP, reducing power, Fe^2+^ chelation	[119]
Mung bean meal	Bromelain, UF	<1 kDa fraction	DPPH, ABTS, SRSA, FRAP, Fe^2+^ chelation	[122]
Mung bean	Trypsin	Hydrolysates	ORAC_,_ ABTS, attenuation of H_2_O_2_-induced oxidative stress in HepG2	[118]
Peeled and split raw Mung bean dry seeds	UF, pepsin, pancreatin, Thermolysin	KK, DM, SY, W (204–274 Da)	ABTS, Fe^2+^ chelation, ORAC	[123]

Abbreviation: NO (nitric oxide).

Legumes such as soybean, lentil, chickpea, and mung bean are familiar sources of food allergens [124]. Enzymatic hydrolysis can reduce or eliminate allergenicity of food proteins, and several proteases have been utilized in the reduction of allergenicity of legumes such as soybean, chickpea, lentil, and mung bean [125,126,127,128]. Interestingly, the mung bean protein hydrolysates produced by a combination of Flavourzyme (nongastrointestinal enzyme) and pancreatin (gastrointestinal enzyme) exhibit potent allergenic activity through the inhibition of β-hexosaminidase release on RBL-2H3 cells [128]. Some peptides that exist in the mung bean protein hydrolysates might potentially serve as a hypoallergenic food formulation or supplement.

## 8. Conclusions

Legume-derived hydrolysates and peptides have attracted interest as functional food ingredients for their applications in managing and reducing the risks of oxidative stress-induced degenerative diseases, inflammatory diseases, hypertension, and cancer. A significant part of the potential health benefits attributed to legumes might be associated with the ability of some amino acids/peptides to scavenge free radicals, inhibit lipid peroxidation, and chelate metal ions. Legume proteins form a significant portion of the diet of people from most regions of the world, and their contribution in terms of health-promoting benefits cannot be ignored. Further, the use of bioinformatics tools is necessary to identify active individual amino acids or peptide sequences for functional food application. Interestingly, some of the legume-derived hydrolysates or peptides exhibit multifunctional properties with health-promoting benefits; therefore, further studies on their wide spectrum of therapeutic action are necessary.

## Figures and Tables

**Table 1 antioxidants-10-00316-t001:** Antioxidant properties of soybean-derived hydrolysates and peptides.

Source of Peptides	Preparation Method/s	Antioxidative Soybean Peptides	Antioxidant Properties (in vitro or in vivo)	Reference
Lunasin	Naturally occurring	SKWQHQQSCRKQLQGVNLTPC, DDDDDDDDD, EKHIMEKIQGRGDDDDDDDDD, EKHIMEKIQ	ABTS, inhibition of ROS generation in LPS-stimulated RAW 264.7 macrophages	[54]
SPI	Alcalase, SEC, IEC	FDPAL (561 Da)	HRSA, SRSA, inhibition of ROS generation in *C. elegans*, increase SOD expression, protection of H_2_O_2_-injured HeLa cells	[52]
Full-fat soybean flakes	Alcalase, pepsin, pancreatin, SEC	GNPDIEHPE, TNDRPSIG, SVIKPPTDE, VIKPPTDE, GNPDIEHPET, LVPPQESQ, EITPEKNPQ, TLVNNDDRDS, NSQHPEL, FEEPQQPQ (823.38–1216.58 Da)	DPPH, ABTS, reducing power, Fe^2+^ chelation, inhibition of ROS generation in Caco-2 cells	[55]
Soy protein fractions	UF, Flavourzyme	<10 kDa	HRSA, ABTS, inhibition of NBT, inhibition of lipid oxidation, FRAP, reducing power	[46]
Black soy sauce	SEC	400–4000 Da (CP1)	ABTS	[56]
Soybean meal protein	Extrusion, Alcalase, UF	Fraction I (>3 kDa), II (3–1 kDa), III (<1 k Da)	HRSA	[47]
Soy industrial effluents	UF, Flavourzyme	Hydrolysates	FRAP, ABTS	[46]
Soybean	Germination, pepsin, pancreatin, UF	Fraction (42.1, 31.4, 19.9, 15.5, 13.6, 12 kDa)	Reducing power, Cu^2+^ and Fe^2+^ chelation, HRSA	[18]
Soybean	UAE	Soybean extracts	ORAC, DPPH	[57]
SPI	Pepsin, papain, chymotrypsin, Alcalase, Protame, Flavourzyme	Hydrolysates	TBARS (liposome system)	[36]
Black soybean soymilk	Pepsin, trypsin	Hydrolysates	FRAP, DPPH	[58]
Soy protein	Alcalase, SEC	Hydrolysates	ABTS	[53]
Soy milk	*Lactobacillus rhamnosus* CRL981	Fermentates	DPPH, inhibition of lipid oxidation, reducing power, inhibition of plasmid DNA oxidation	[49]
Soy milk	*Lactobacillus acidophilus* (*CCRC 14079* or *Streptococcus thermophilus CCRC 14085*), and *Bifidobacterium infantis* (*CCRC 14633* or *Bifidobacterium longum B6*)	Fermentates	Inhibition of ascorbate autoxidation, reducing activity, SRSA	[50]
Soy milk	*Lactobacillus rhamnosus strains (L. rhamnosus C8, L. rhamnosus C25, L. rhamnosus C28,* or *L. rhamnosus C34*)	Fermentates	ABTS, DPPH, HRSA	[59]
Soy milk	*Leuconostoc sp. MYU 51, Lactobacillus sakei MYU 57, Leuconostoc mesenteroides* MYU 60, *Lactobacillus sakei* MYU 67, *Lactobacillus gasseri* MYU 1, *Pediococcus pentosaceus* MYU 759	Soy yogurt, soymilk yogurt supernatants	ORAC, HORAC, inhibition of ROS generation in HCT 116 cells, DNA protection (comet assay)	[60]
Soy milk fortified with 2% whey protein concentrate	*Lactobacillus rhamnosus* (NCDC 17, 19, 24, 297, C2, C6)	Fermentates	ABTS, DPPH, FRAP	[61]
Soy milk	*Bacillus subtilis* (2805PNU014, 2829PNU015, 2825PNU016, MYCO10001, KCCM11316)	Fermentates	DPPH, ABTS, LDL oxidation inhibition	[62]
Soy milk	*Lactobacillus plantarum*	Fermentates	DPPH, HRSA, inhibition of ROS generation and lipid peroxidation in Caco-2 cells, enhanced levels of CAT, SOD, GSH-Px	[63]
Soybean meal extract	*Lactobacillus plantarum* strain RM10	Fermentates	In vitro: DPPH, Fe^2+^ chelation In vivo: inflammation/infection model (Wistar rat; MDA, MPO, GSH levels)	[41]
SPI	*Chryseobacterium* sp. kr6	Fermentates	ABTS, DPPH, Fe^2+^ chelation	[12]
SPI	Germination	Germinates	DPPH, ABTS, HRSA	[58]
SPI	Heating, proteolysis, glycation, UF	Hydrolysates	LDL oxidation inhibition, ORAC	[64]

Abbreviations: ABTS (2,2′-azino-bis (3-ethylbenzothiazoline-6-sulfonic acid), ROS (reactive oxygen species), LPS (lipopolysaccharide), SPI (soy protein isolate), SEC (size exclusion chromatography), IEC (ion-exchange chromatography), HRSA (hydroxy radical scavenging activity), SRSA (superoxide radical scavenging activity), SOD (superoxide dismutase), DPPH (2,2-diphenyl-1-picrylhydrazyl), UF (ultrafiltration), NBT (Nitroblue tetrazolium), RAP (ferric-reducing antioxidant power), GSH-Px (glutathione peroxidase), MDA (malondialdehyde), UAE (ultrasonic-assisted extraction), ORAC (oxygen radical absorbance capacity), TBARS (thiobarbituric acid reactive substances), HORAC (hydroxyl radical antioxidant capacity), LDL (low-density lipoprotein), CAT (catalase), MPO (myeloperoxidase), GSH (glutathione).

**Table 2 antioxidants-10-00316-t002:** Antioxidant properties of chickpea-derived hydrolysates and peptides.

Source of Peptides	Preparation Method/s	Antioxidative Chickpea Peptides	Antioxidant Properties (in vitro or in vivo)	Reference
Chickpea protein	Alcalase, SEC	NFYHE (717.37 Da)	DPPH, HRSA, SRSA, Cu^2+^ and Fe^2+^ chelation, inhibition of linoleic acid autoxidation	[34]
Chickpea protein	Alcalase, SEC	Low molecular weight peptides (two peaks of 940–2622 Da and 220–940 Da)	Reducing power, DPPH, MRSA, SRSA, inhibition of linoleic acid autoxidation	[80]
Chickpea protein	Pepsin, pancreatin, affinity chromatography with immobilized copper, SEC, HPLC	Low molecular weight peptides (105–1205 Da)	Cu^2+^ chelation	[83]
Chickpea protein isolates	RP-HPLC	ALEPDHR (836.2 Da), TETWNPNHPEL (1336.3 Da), FVPH (498.1 Da), SAEHGSLH (836.1 Da)	Reducing power, FRSA, inhibition of peroxyl-induced oxidation in Caco-2 cells	[32]
Chickpea protein	Alcalase, Flavourzyme, SEC	RQSHFANAQP (1155 Da)	DPPH, ABTS, HRSA	[33]
Chickpea protein concentrate	Alcalase, SEC, RP-HPLC	DHG (327.33 Da), VGDI (402.49 Da)	DPPH, reducing power, inhibition of lipid oxidation, Fe^2+^ chelation	[74]
Chickpea sprout protein	Trypsin, Neutrase, Alcalase, papain, IEC, SEC, RP-HPLC	LTEIIP (685.41 Da)	DPPH, HRSA	[88]
Chickpea seeds protein	Alcalase, Flavourzyme	Hydrolysates	DPPH, inhibition of lipid oxidation	[75]
Chickpea	Germination	Low molecular weight peptides (9.4–10.9 kDa)	DPPH	[43]
Chickpea seeds	Pepsin, pancreatin, UF	Low molecular weight peptides (3.5–7 kDa)	ABTS, DPPH, Fe^2+^ and Cu^2+^ chelation	[90]

Abbreviations: HPLC (high-performance liquid chromatography), RP-HPLC (reverse-phase high performance liquid chromatography).

**Table 3 antioxidants-10-00316-t003:** Antioxidant properties of hydrolysates and peptides from lentil protein.

Source of Peptides	Preparation Method/s	Antioxidative Chickpea Peptides	Antioxidant Properties (in vitro or in vivo)	Reference
Lentil seeds	Germination	Germinates	DPPH, reducing power	[44]
Lentil protein	Savinase, UF	LLSGTQNQPSFLSGF (1595.81 Da), NSLTLPILRYL (1302.78 Da), TLEPNSVFLPVLLH (1578.89 Da)	ORAC	[95]
Lentil seeds	Hydrostatic pressure, Alcalase, Protamex, Savinase, Corolase 7089	<3 kDa peptide	ORAC	[94]
Lentil seeds	*Lactobacillus plantarum*, *Bacillus subtilis*	Fermentates	ORAC	[92]
Lentil	*Aspergillus oryzae*, *Aspergillus niger*	Fermentates	DPPH, ABTS, FRAP	[91]
Lentil	*Lactobacillus plantarum* CECT 748, Savinase, SEC	SDQENPFIFK (1225 Da), HGDPEER (839 Da), ATAFGLMK (838 Da)	ORAC, inhibition of ROS generation in RAW 264.7 macrophages	[93]

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
