# Peer review of "Health Benefits of Antioxidative Peptides Derived from Legume Proteins with a High Amino Acid Score"

_antioxidants, 2021, doi:10.3390/antiox10020316_

Round 1
Reviewer 1 Report
The main aim of this article was to review the potential antioxidative properties of legume-derived peptides, focusing on their in vitro and in vivo effects. The topic tackled by this manuscript is timely and addresses and important issue in the functional foods and specifically that related with antioxidant properties. The scientific content of the paper is good and represents a real advance in the knowledge of the area it deals. Moreover, it is clearly written showing numerous tables (six tables) and citing many bibliographic references consulted (143 references). Therefore, I recommend its acceptation after minor revision.
I would like to make the following suggestions:
- Keywords: Health benefits and amino acid should be added.
- Table 1: It should appear just it has been cited into the text.
- Table 2 and corresponding text: The peptide Phe-Asp-Pro-Ala-Leu should be written such as FDPAL.
- Table 3 and corresponding text: The peptides should be written only with the nomenclature of a single letter.
- References: Some mistakes should be corrected. For example, in reference 1 the authors should be cited such as Garcia-Nebot M.J.; Recio I., Hernández-Ledesma B., in reference 12 such as Lorenzo J.M.; Munekata P.E.S.; Gómez B.; Barba F.J.; Pérez-Santaescolástica C.; Toldrá F, in reference 14 such as De Mejia E.G; De Lumen B.O, reference 16 such as González-Montoya M.; Ramón-Gallegos E.; Robles-Ramírez M C.; Mora-Escobedo R, in reference 40 such as Hernández-Ledesma B.; Hsieh C.; De Lumen B.O, in reference 61 such as Mesa M.D.; Silván J.M.; Olza J.; Gil Á.; Del Castillo M.D, in reference 75 such as Wang W.; De Mejia E.G; in reference 80 such as Wenyi W.; De Mejia, E.G., in reference 84 such as Domínguez-Vega E.; Kotkowska O.; García M.C.; Crego A.L.; Marina M.L; in reference 107 such as Alves Magro A.E; Silva L.C.; Rasera G.B.; Soares de Castro R.J; in reference 108 such as Torino M.I.; ; Limón R.I.; Martínez-Villaluenga C., et al.; in reference 109 such as Bautista-Expósito S.; Martínez-Villaluenga C.; Dueñas, M.; Silván J.M; Frias J.; Peñas E.; in reference 111 such as García-Mora P.; Martín-Martínez M.; Bonache M.A., et al.; in reference 120 such as Barnes M.J.; Uruakpa F.O.; Udenigwe, C.C, in reference 121 such as Rocha T.D.S.; Hernandez L.M.R.; Chang Y.K.; De Mejia E.G. Moreover, some references look incomplete. i.e references 2, 3, 142 and 143. The names of the manuscript should be written without capital letters i.e. references 2, 3, 6, 7, 13, 16, 17, 27, 31, 38, 39, 41, 45, 47, 48, 50, 51, 58, 62, 67, 68, 70, 77, 78, 80, 97, 98, 100, 101, 125, 126, 131, 133, and 143. However, the initials of the abbreviated journal name should be written in capital letter i.e. Int J Food Sci Technol in reference 49 and, Eur J Lipid Sci Technol in reference 76.
- Some mistakes:
- Lane 577: ace should be changed by ACE.
Author Response
We are grateful to the critical comments that have helped us to improve the manuscript considerably. According to reviewer’s comments, we have carefully revised the manuscript. The revised parts are highlighted in red in the manuscript. The responses to reviewer’s and editor’s comments are as follows.
Keywords: Health benefits and amino acid should be added.
Response: As your suggestion, “Health benefits” and “amino acids” were added (lines 22-23).
Table 1: It should appear just it has been cited into the text.
Response: Thank you for your point out. As the other reviewer suggested, we deleted Table 1.
Table 2 and corresponding text: The peptide Phe-Asp-Pro-Ala-Leu should be written such as FDPAL.
Response: We revised it to FDPAL. The nomenclature of amino acid in all peptide was presented as a single letter. In the text, we revised in the same way.
Table 3 and corresponding text: The peptides should be written only with the nomenclature of a single letter.
Response: Thank you for your point out. The amino acids residues in peptide sequences were revised to single letter.
References: Some mistakes should be corrected.
Response: Thank you for pointing out concerning the reference format. All the references were corrected according to your suggestion (lines 404-674).
Lane 577: ace should be changed by ACE.
Response: Since the title of this reference paper is “ace”, we use the word “ace” according to the original (line 567).
Reviewer 2 Report
I have the following comments and suggestions:
- The Abstract is too short.
- How appropriate references were found should be noted in the abstract.
- There are missing part in the Manuscript about references findings. It must be added.
- It would be good if certain meta analysis is performed.
- Adverse effects of allergenic legumes' properties should be mentioned too. The following reference can be used:Kalčáková, L., Tremlová, B., Pospiech, M., Hostovský, M., Dordević, D., Javůrková, Z., ... & Bartlová, M. (2020). Use of IHF-QD Microscopic Analysis for the Detection of Food Allergenic Components: Peanuts and Wheat Protein. Foods, 9(2), 239.
Author Response
We are grateful to the critical comments that have helped us to improve the manuscript considerably. According to reviewer’s comments, we have carefully revised the manuscript. The revised parts are highlighted in red in the manuscript. The responses to reviewer’s and editor’s comments are as follows.
The Abstract is too short.
Response: According to the suggestion, more specified explanation was added in the abstract (lines 16-18).
How appropriate references were found should be noted in the abstract.
Response: Thank you for your suggestion. Since we don’t use special method, we avoid to explain it.
There are missing part in the Manuscript about references findings. It must be added.
Response: As the reviewer mentioned, we carefully revised the missing reference in the Manuscript.
It would be good if certain meta-analysis is performed.
Response: We agree with the suggestion. This work focuses on lots of works concerning antioxidant activity of legume peptides, and meta-analysis is a powerful tool. Authors will consider performing meta-analysis in other coming reviews.
Adverse effects of allergenic legumes' properties should be mentioned too. The following reference can be used: Kalčáková, L., Tremlová, B., Pospiech, M., Hostovský, M., Dordević, D., Javůrková, Z., & Bartlová, M. (2020). Use of IHF-QD Microscopic Analysis for the Detection of Food Allergenic Components: Peanuts and Wheat Protein. Foods, 9(2), 239
Response: Authors are thankful for sharing the article. The explanation regarding allergenicity of food legumes has been added (lines 376-384).

Reviewer 3 Report
The manuscript “Health Benefits of Antioxidative peptides Derived from Legume Proteins with a High Amino Acid Score” cites the health benefits of legume-based proteins in general and biofunctional peptides in particular. These peptides can have health benefits when consumed and/or used in creating functional foods. The authors identify and characterize known peptides from some commonly used legumes, describing amino acid content, health effects, and antioxidant capacity. The list of references is impressive and contains much information about the characteristics of these peptides.
The large amount of information could be helpful as a resource. Its value would be enhanced by more explicitly highlighting the value and possible application of the data. The authors do state in lines 50-53 that identifying effective peptides can help to establish structure activity relationships, but this assertion would be strengthened by more clearly stated conclusions based on the data they present.
Specific comments and questions
- The authors might consider clarifying the focus of the paper as reported in the title: is it reporting of health effects of antioxidant peptides or reporting antioxidant effects of healthful peptides?
- Line 41 Consider replacing ref 3, which appears to be an unpublished thesis, with other references. Possible suggestions: Caleja C, Barros L, Antonio AL, Oliveira MB, Ferreira IC. A comparative study between natural and synthetic antioxidants: Evaluation of their performance after incorporation into biscuits. Food Chem. 2017 Feb 1;216:342-6. Lourenço SC, Moldão-Martins M, Alves VD. Antioxidants of Natural Plant Origins: From Sources to Food Industry Applications. Molecules. 2019;24(22):4132. Published 2019 Nov 15. doi:10.3390/molecules24224132
- Lines 36-38. Are the peptides described produced during consumption of whole legumes or are they only produced chemically? If they are not present in the body on consumption of legumes, what is their relevance to health since many studies of the health effects of legumes do examine the whole food (see lines 166-170), and legume consumption is likely to be more common than consumption of specific peptides? It is true though that health effects of some peptides are cited.
- Table 1. What were the criteria for choosing the five legumes in Table 1?
- Lines 67-112. This section could be framed more clearly. Plant proteins are described as being incomplete, especially deficient in sulfur-containing amino acids that have high antioxidant capacity (lines 75-76). Lines 92-101 describe Tyr, Cys, Met, His, and Trp as having antioxidant activity, but these seem to be described as limiting amino acids in Table 1. Although the legume proteins do have overall a sufficiently high amino acid score, the relative lack of antioxidant amino acids does not seem to suggest high antioxidant activity.
- Line 287. Is HDL or LDL meant here? Lowering HDL would be deleterious.
7. It would be helpful to see more of a summary of the data in the conclusions (lines 391-400) which is somewhat general. The authors comment throughout the text on structural characteristics vs antioxidant activity (e.g., line 244-249) or antioxidant activity vs reported health effects (lines 149-151) but it would be interesting to see a final analysis of their data in this light.
Author Response
We are grateful to the critical comments that have helped us to improve the manuscript considerably. According to reviewer’s comments, we have carefully revised the manuscript. The revised parts are highlighted in red in the manuscript. The responses to reviewer’s and editor’s comments are as follows.
The large amount of information could be helpful as a resource. Its value would be enhanced by more explicitly highlighting the value and possible application of the data. The authors do state in lines 50-53 that identifying effective peptides can help to establish structure activity relationships, but this assertion would be strengthened by more clearly stated conclusions based on the data they present.
Response: We thank the reviewer’s suggestion. As the reviewer mentioned, this paragraph was not suitable. Therefore, we reframed this paragraph and the next paragraph (lines 57-64)
The authors might consider clarifying the focus of the paper as reported in the title: is it reporting of health effects of antioxidant peptides or reporting antioxidant effects of healthful peptides?
Response: The focus of the paper is to comprehensively review the health effects of legume-protein derived antioxidant peptides, and thus we use the original title.
Line 41 Consider replacing ref 3, which appears to be an unpublished thesis, with other references. Possible suggestions: Caleja C, Barros L, Antonio AL, Oliveira MB, Ferreira IC. A comparative study between natural and synthetic antioxidants: Evaluation of their performance after incorporation into biscuits. Food Chem. 2017 Feb 1; 216:342-6. Lourenço SC, Moldão-Martins M, Alves VD. Antioxidants of Natural Plant Origins: From Sources to Food Industry Applications. Molecules. 2019; 24(22):4132. Published 2019 Nov 15. doi:10.3390/molecules24224132
Response: The authors are thankful for the proposed articles. “Bissegger, 2008” was replaced by “Lourenço et al., 2019”, ref 5 (line 50 and lines 410-411).
Lines 36-38. Are the peptides described produced during consumption of whole legumes or are they only produced chemically? If they are not present in the body on consumption of legumes, what is their relevance to health since many studies of the health effects of legumes do examine the whole food (see lines 166-170), and legume consumption is likely to be more common than consumption of specific peptides? It is true though that health effects of some peptides are cited.
Response: We thank for the concrete point out. Actually, many biofunctional peptides can be produced by digestive enzymes during gastrointestinal digestion, and fermentation also produce bioactive peptides. To clarify the importance of gastrointestinal digestion and fermentation, we revised some sentences (lines 43-45).
Table 1. What were the criteria for choosing the five legumes in Table 1?
Response: According to the reviewer’s suggestion, we have added the explanation concerning the criteria for choosing five legumes (lines 26-30).
Lines 67-112. This section could be framed more clearly. Plant proteins are described as being incomplete, especially deficient in sulfur-containing amino acids that have high antioxidant capacity (lines 75-76).
Response: According to the reviewer’s suggestion, we have added the explanation concerning the high antioxidant capacity of lentil and mug bean regardless of lower sulfur-containing amino acids (lines 68-75).
Lines 92-101 describe Tyr, Cys, Met, His, and Trp as having antioxidant activity, but these seem to be described as limiting amino acids in Table 1. Although the legume proteins do have overall a sufficiently high amino acid score, the relative lack of antioxidant amino acids does not seem to suggest high antioxidant activity
Response: We thank for a concrete suggestion about the limiting amino acids. We revised the sentences concerning limiting amino acids (lines 68-75). Since Table 1 contains confusing data concerning limiting amino acids, we deleted Table 1.
Line 287. Is HDL or LDL meant here? Lowering HDL would be deleterious.
Response: “LDL” is wrong, and the correct is “HDL”. We deleted “HDL cholesterol” because we didn’t give mention concerning cholesterol in this paragraph (line 272).
It would be helpful to see more of a summary of the data in the conclusions (lines 391-400) which is somewhat general. The authors comment throughout the text on structural characteristics vs antioxidant activity (e.g., line 244-249) or antioxidant activity vs reported health effects (lines 149-151) but it would be interesting to see a final analysis of their data in this light.
Response: We thank the reviewer’s suggestion. the explanation concerning more specified conclusion was added (lines 386-391).

Round 2
Reviewer 2 Report
It can be accepted.
Reviewer 3 Report
The revisions are helpful in clarifying the work.